# Identifying the Strength Level of Objects’ Tactile Attributes Using a Multi-Scale Convolutional Neural Network

**DOI:** 10.3390/s22051908

**Published:** 2022-03-01

**Authors:** Peng Zhang, Guoqi Yu, Dongri Shan, Zhenxue Chen, Xiaofang Wang

**Affiliations:** 1School of Electrical Engineering and Automation, Qilu University of Technology (Shandong Academy of Sciences), Jinan 250353, China; wxf2012@stu.xjtu.edu.cn; 2School of Mechanical Engineering, Qilu University of Technology (Shandong Academy of Sciences), Jinan 250353, China; 1043119037@stu.qlu.edu.cn (G.Y.); shandongri@qlu.edu.cn (D.S.); 3School of Control Science and Engineering, Shandong University, Jinan 250061, China; chenzhenxue@sdu.edu.cn

**Keywords:** robot tactile, convolution neural network, attribute strength level, robot operating system

## Abstract

In order to solve the problem in which most currently existing research focuses on the binary tactile attributes of objects and ignores identifying the strength level of tactile attributes, this paper establishes a tactile data set of the strength level of objects’ elasticity and hardness attributes to make up for the lack of relevant data, and proposes a multi-scale convolutional neural network to identify the strength level of object attributes. The network recognizes the different attributes and identifies differences in the strength level of the same object attributes by fusing the original features, i.e., the single-channel features and multi-channel features of the data. A variety of evaluation methods were used for comparison with multiple models in terms of strength levels of elasticity and hardness. The results show that our network has a more significant effect in accuracy. In the prediction results of the positive examples in the predicted value, the true value has a higher proportion of positive examples, that is, the precision is better. The prediction effect for the positive examples in the true value is better, that is, the recall is better. Finally, the recognition rate for all classes is higher in terms of f1_score. For the overall sample, the prediction of the multi-scale convolutional neural network has a higher recognition rate and the network’s ability to recognize each strength level is more stable.

## 1. Introduction

The diversity of objects is reflected in the different attributes of the objects and the difference in the strength level of the same attributes. When modeling the diversity of objects based on deep learning methods, such as the image classification of visual data [1] and the speech recognition of audio data [2], it is often necessary to pay attention to the differences among objects. In the field of haptics, the difference in haptic signals often reflects the difference in the properties of measured objects, which is an important basis for classifying the input into discrete categories.

In recent years, with the development and application of deep learning theory, many scholars have taken into account the unique advantages of tactile information and have begun to devote themselves to the fields of tactile perception and attribute recognition by using deep learning methods [3]. The tactile sensor can provide the robot with feedback information, such as the interaction force [4]: for example, whether or not the sensor is in contact with the object [5], whether there is sliding [6], and what the physical properties of the object are, such as its temperature [7], roughness, and texture [8,9].

Because objects with different material properties deliver different feedback to the tactile sensor during the interaction, the task of many deep learning methods is to classify tactile data into discrete categories to identify objects [10,11]. Robots can collect this feedback information through dynamic exploration programs [12,13], for example, [14,15] letting volunteers touch objects to feel their physical characteristics to learn the meaning of tactile attribute adjectives, and establishing a tactile data set by designing an exploration program, but the authors only require each volunteer to give a binary label (yes/no). The data set cannot evaluate the attributed strength level of objects. In addition, many scholars have conducted in-depth research on the identification of the object’s temperature, surface texture, softness, and hardness. For example, Ref. [14] used a convolutional neural network (CNN) to analyze the time series data of pressure and six-axis acceleration sensors to obtain tactile texture information. The static friction coefficient of an object can be estimated by measuring the normal force and tangential force during the initial slip [16]. In [17], a variety of deep learning models are used to explore surface roughness, and the feature extraction capabilities of models with different roughness ranges are introduced in detail. In [18], an artificial nail with a three-axis acceleration sensor is used to recognize the surface of vibration when scratching the surface of different materials, and proves that good performance can be obtained by using a variety of exploration behaviors. In [19], according to the signal amplitude caused by vibration, the k-nearest neighbor method is used to distinguish different surfaces. In [20], a CNN is used to identify the tactile information of human or non-human objects in disaster scenes to protect people. In [21], a CNN is used to predict object attributes, but the feature extraction of each attribute requires a classifier, and the amount of network parameters is large.

Many researchers try to teach robots to recognize adjectives concerning an object’s tactile attributes from raw data, try to mark objects by using adjectives concerning binary tactile attributes (for example, hard or not hard), and map the characteristics of tactile data. However, the above method has the following shortcomings:In the existing public data set [14,15], experimenters are only required to give each object attribute a binary label (yes/no). The binary label does not include the knowledge of the strength level of the object’s delicate attribute.In real scenes, humans who have a rich tactile perception system will obtain a delicate sensory feedback by touching an object and will have a specific cognition of the strength level of the object’s attribute. For networks, only using binary tactile labels to describe objects will simplify object attributes to binary space and the network has a very rough understanding of the level of the strength of an object’s attribute [22].

In order to solve the above problems, first of all, this paper established a tactile-based object attribute data set. Samples in the data set contain the elasticity and hardness characteristic information of objects, and each attribute is divided into 10 levels to describe the object more delicately. Compared with other tactile data sets in this field, the unique contribution of the tactile data set established in this paper is to solve the problem of lack of relevant data about the intensity difference of object attributes and to add information about the intensity difference of object attributes in the data set. The data set lays a foundation for more refined cognition and operation of objects through robot touch. Second, a method for identifying the attribute strength level of a convolutional neural network by using multi-scale features is proposed, which can identify the difference in the strength of elasticity and hardness of different objects in the data set. The main contribution of this method is that, in each feature extraction process, the multi-scale features of the original data are comprehensively superimposed, which makes the network understand the data more comprehensively. Compared with other methods, the unique advantage of this method is that the integrity and sufficiency of information are fully considered in each step of feature extraction, which means that this method is sensitive to data changes. The elasticity strength is regarded as the strength of the feedback to the experimenter when the object returns to its original state after deformation, and the hardness is regarded as the difficulty of the deformation of the object under the same force.

The structure of this paper is as follows: in Section 2, the latest developments related to this topic are introduced. In Section 3, the establishment process of the tactile data set is introduced. Section 4 introduces the recognition algorithm for the strength level of object attributes of a multi-scale convolutional neural network. Section 5 introduces the experimental results of the recognition algorithm on the tactile data set. Finally, Section 6 provides the discussion and Section 7 provides the conclusion.

## 2. Related Work

The robot’s tactile object recognition capability is achieved by processing haptic signals, which can be represented as a continuous signal, a set of discrete measurements, or a series of images [20]. How the data are processed may be affected by different data structures. This paper reviews the related studies on the use of haptic data to identify their attributes from the perspective of the data structure of haptic signals.

At present, the common haptic signals are force signals and vibration signals. If tactile sensors provide a pressure value, the best form of haptic signals may be a time-varying pressure curve, which can be used to detect contact or sliding events between contact surfaces [23,24]. Some studies consider the static information of the pressure images [25]. Although the object is pressed several times, each pressure image only contains the pressure distribution information caused by the shape of the object and does not contain the time relationship between the images [26]. Therefore, some studies have represented each image as a matrix of pressure readings at an instant, which can contain the physical properties of an object’s changes in information over time. For a haptic image, machine learning methods, such as the k-nearest neighbor method [19], Bayesian method [27], and the traditional method based on images [28,29], were used to identify features. However, this did not mean that these corresponding methods must be used in the extraction process of the haptic feature.

There has been some progress in deep learning using haptic data to identify objects. In [30,31], a tactile recognition system using a deep learning method was proposed. This system can recognize objects by grasping objects, but it cannot recognize the physical properties of objects. In [32], a deep learning multi-class and multi-label model was designed. The model can identify targets by learning four tactile features, including hardness, thermal conductivity, roughness, and texture from haptic images. Considering the good performance of convolutional neural network processing to extract spatial features, the use of convolutional neural networks to process tactile images is widely used [21,33]. In addition, some researchers use the advanced processing of haptic pressure images to classify and recognize objects [34]. For example, the pressure images obtained during extrusion and release are connected into a tensor that can be used to classify objects in 3D CNN.

Regardless of the form of the data structure of the haptic signal, one way to obtain richer information about the haptic signal is for humans to classify haptic samples using discrete categories that are more detailed than a binary label. The simplest partitioning task is to classify objects according to their similarity and select one or more dimensions for analysis. In [35], the results of the free sorting of different material samples were analyzed by multidimensional scaling, and the tactile material space was calibrated by the physical measurements of compressibility and roughness. Similarly, Ref. [36] discussed the main dimensions of tactile surface perception. Roughness, smoothness, and softness proved to be important orthogonal dimensions, and it was concluded that elasticity may correspond to the third main dimension. However, in a later study [37], the third major dimension of tactile perception was identified as viscosity or slippage. In addition, in [14,15], a number of different researchers were selected to give binary labels to adjectives of different objects, but the antisense relationships between adjectives were not taken into account. Therefore, Ref. [22] complements this aspect of the study by confirming the antonym pairs of hard/soft, rough/smooth, and cold/warm, and more tactile information than the binary label is analyzed.

## 3. Materials

The haptic data sets Penn Haptic Adjective Corpus-1 (PHAC-1) and Penn Haptic Adjective Corpus-2 (PHAC-2) are proposed in [14,15]. However, each volunteer was only required to give the binary label (yes/no) of each object attribute, but the binary label cannot judge the strength of the object attribute. Therefore, this article uses its own tactile data collection platform to establish a tactile data set of the strength levels of the object.

### 3.1. Robot Platform

As shown in Figure 1, the desktop Kinova robotic arm equipped with SynTouch’s NumaTac tactile sensor is selected as the haptic data acquisition platform in this paper.

The Kinova arm has seven degrees of freedom, including a manipulator and a two-finger gripper. Two NumaTac tactile sensors are mounted on each finger of the two-finger gripper. The original data collected include the DC pressure signal (*P_DC_*) at a sampling frequency of 100 Hz and the AC pressure vibration signal (*P_AC_*) at a sampling frequency of 2200 Hz. The DC pressure value and AC pressure value per unit area can be calculated with Formulas (1) and (2):DC = (*P_DC_* − offset) × 12.94 Pa/bit(1)
AC = (*P_AC_* − offset) × 0.13 Pa/bit(2)

In Formula (1), offset is the DC pressure signal value of the NumaTac tactile sensor signal under atmospheric pressure, and *P_DC_* is the instantaneous DC pressure signal value obtained by the NumaTac tactile sensor when the two-finger gripper interacts with the object. The resolution of 12.94 is obtained from the official documentation of the NumaTac tactile sensor. In Formula (2), offset is the AC pressure vibration signal value of the NumaTac tactile sensor under atmospheric pressure. *P_AC_* is the instantaneous AC pressure vibration signal value obtained by the NumaTac tactile sensor when the two-finger gripper interacts with the object, and the resolution of 0.13 is obtained from the official document of the NumaTac tactile sensor. The rest of the relevant information about the NumaTac tactile sensor can be found on SynTouch’s official website.

The tactile data acquisition platform uses the robot operating system (ROS) as the software interface. ROS has a series of libraries and tools to help developers write robot software programs [38], which is an ROS node. When a complex task is completed through a series of programs, ROS creates a network that connects all nodes, which is an ROS graph. Through the interaction between the ROS graph, other nodes can obtain the information published.

In this paper, the nodes included in the ROS graph mainly include the Kinova node, the SynTouch node, and the process node. The registration, communication, and parameter server of the nodes are managed by the ROS master, as shown in Figure 2.

The ROS master is the core of ROS. It registers the names of nodes, services, and topics and maintains a parameter server.The Kinova node plans a reasonable path to move the end of the manipulator to a fixed point by manipulating the Kinova manipulator, then manipulating the gripper to open and close the object. The gripper continues to close after contacting the object. The gripper stops closing when the fixed force threshold is reached.The SynTouch node is used to publish the tactile data generated by the interaction between the tactile sensor and the object. During the execution of a complete exploration action to generate tactile data, the two tactile sensors following the two-finger gripper physically interact with the object to obtain the tactile sensor data and publish data to the ROS network at a frequency of 100 Hz.The judge is used to judge whether the tactile sensor is in contact with the target object. When the ratio of the *P_AC_* signal value
$value_{t+1}$ at the next moment to $value_{t}$ at the previous moment is greater than 1.1 or less than 0.9, it is considered that the initial touch occurs and the data will be published to the process node. The values 1.1 and 0.9 are experimentally measured thresholds that can determine whether the initial contact is generated.The process node subscribes to the haptic data released by the SynTouch node and judges whether an initial contact occurs. After satisfying the conditions, it intercepts the data released by the SynTouch node and superimposes them into a dual-channel haptic sample. Since the Kinova robotic arm needs to reach the preset position before performing the exploration action, the NumaTac returns useless data in this process. In order to solve this problem, a sub-thread is established. First, it continuously receives tactile data in the main thread and judges whether the tactile sensor has initial contact with the object. When the initial contact is made, the sub-thread waits for 3 s until all the sample data in the main thread are received, where 3 is the experimentally measured time required to collect a complete sample. The data are then intercepted to create haptic samples.

### 3.2. NumaTac Haptic Dataset

This paper uses the abovementioned robot platform to physically interact with some common objects in daily life repeatedly in order to collect a large number of tactile samples. Then, we let four experimenters physically interact with the objects to provide the elasticity strength level and hardness strength level of each object as the object labels. The elasticity and hardness properties of each object are coupled together. These constitute the NumaTac haptic data set.

#### 3.2.1. Objects

Considering the foam material on the surface of the NumaTac tactile sensor and the movement limitation of the Kinova, 30 different objects were selected after excluding the objects with dangerous characteristics, such as sharpness, high temperature, humidity, and unsuitable scale. Objects contain a variety of material properties representing a wide range of physical properties, as shown in Figure 3. The 30 objects contain a variety of shapes and materials, which can meet our needs to realize attribute strength difference recognition.

#### 3.2.2. Data Collection Detail Settings

Humans can use a range of grasping movements when evaluating objects. Similarly, the motions of the Kinova to grasp objects in this paper are identified as squeeze, static grip, and release. Since the interaction between the Kinova and the object is a continuous process, the above Kinova actions are combined into a complete process to collect tactile data. The specific ROS graph is shown in Figure 2.

Kinova performs the abovementioned grasping action to obtain the haptic data of the object. By adjusting the position of the object, the object is kept on the mid-axis plane of the two fingers, and the slight posture change is set to simulate the uncertainty of touch. When the squeezing action occurs, the two fingers touch the object almost at the same time, preventing the object position offset from interfering with the two tactile sensors and reducing unnecessary changes between experiments. 

During the gripping process, Kinova remains in a fixed position, and the whole process is just one grasping motion. The tiny robotic arm’s jitter information is ignored and only the signal generated by the tactile sensor is considered. When the grasping action begins, the gripper closes at a constant speed until the NumaTac makes initial contact with the object. At the beginning of the squeeze phase, the NumaTac continuously deforms to obtain tactile signals until it reaches the force threshold of 3493.8 pa, and the *P_DC_* value of one NumaTac reaches 270 bit (the offset is 238). The callback function in the node sends a stop command to the gripper. Since the signal transmission takes time, the gripper will continue to squeeze the object for a short period of time before receiving the stop command, and will continuously send data back. When the stage of squeezing the object ends, the gripper enters the stage of static gripping the object, the deformation of the NumaTac remains unchanged, and the tactile signal is continuously received for a period of time after the static grip phase is over. Then, the gripper opens at a constant speed and enters the release phase, where the NumaTac restores its initial shape and state and generates a tactile signal.

A total of 30 objects were grasped and each object was grasped 50 times by performing the above continuous grasping motion using a tactile data acquisition platform. All *P_DC_* and *P_AC_* were recorded.

#### 3.2.3. Data Preprocessing

The time required for a complete data collection action is 3 s, and the frequency at which the NumaTac haptic data are returned to the ROS network is 100 Hz. Therefore, the length of each sample data is set to 300 discrete data points. Since the process is a continuous process, all initial samples are more than 300 discrete points in length. For samples with more than 300 discrete points, appropriate data segments are intercepted.

Considering the symmetry of the mechanical structure when the robot’s dexterous hand grasps the object and the integrity of the tactile information of the measured object, we increased the feature quantity of the sample’s tactile information, and superimposed the tactile data of the left and right fingers of each sample into a dual-channel sample.
(3)$$x^{*}=\frac{x-\mu }{\sigma }$$

In Formula (3), $x^{*}$ is the normalized data, x is the original data, μ is the mean over the feature dimension of x, and σ is the standard deviation in the feature dimension of x. In order to eliminate the initial value of the influence of atmospheric pressure on the tactile sensor, the normalization function in Formula (3) is used to eliminate the influence of the initial value of the tactile sensor when there is no object contact.

#### 3.2.4. Labels

To match an object’s haptic data with its own real-world label, four experimenters were given eye patches and headphones, and they were instructed to use two fingers to simulate the grasping action of a gripper touching the object and compare the strength levels of the object’s properties. All experimenters were asked to evaluate the elasticity strength level and the hardness strength level separately for each object, and the levels were limited to 10 levels from 1 to 10. The label dimension of each object is 1 × 20. The first 10 grades represent elasticity, and the last 10 grades represent hardness. For example, the label of a mineral water bottle is (0,0,0,0,0,0,0,0,1,0,0,0,1,0,0,0,0,0,0,0) in Figure 4.

Regarding the question of whether the data set is balanced, since the focus is on the properties of the elasticity and hardness of the object, each sample contains these two kinds of information. The data set has the same sample size of each strength level of objects’ tactile attributes, and each strength level has 150 samples, so the data set is a balanced data set.

For the problem of different strength levels given by different experimenters, the experimenters can vote on all the strength levels given, and the experimenters could agree with their own or others’ judgment of the object. This process is completely decided by the internal discussion of the four experimenters, without the influence of other external personnel. The experimenters are required to give unique elasticity and hardness strength levels for each object that will not be changed.

## 4. Method

### 4.1. Multi-Scale Convolutional Neural Network Architecture

The recognition of the tactile attribute strength level of objects is an important issue. One way to obtain rich tactile information with human perception is to use more discretized categories to represent the similarity of object attributes and the difference of attribute strength levels. The feature extraction of tactile information is the basis of object attribute strength level recognition. Considering the good ability of CNN feature extraction, this work used a multi-scale convolutional neural network to extract the tactile information features of objects. The network structure is shown in Figure 5.

The multi-scale convolutional neural network consists of an input layer, a block module, and a fully connected module. The block feature extraction module contains three scales: the original feature map, single-channel feature map, and multi-channel feature map.

The original feature map: the obtained feature map is subjected to a max-pooling operation with a kernel size of two to correspond to the size of the other two convolutional layers, and no convolution operation is performed to preserve the original information.The single-channel feature map: the output of the previous step is subjected to a convolution operation through conv1_1, conv2_1, and conv3_1, and a max-pooling operation with a kernel size of two is performed, and then multiplied by a weight coefficient
$k^{1}$ of a convolutional layer. The $k^{1}$ is used for the weight coefficient of training single-channel output features in the network.The multi-channel feature map: the output feature map of the previous step is subjected to a convolution operation through conv1_2, conv2_2, and conv3_2, and a max-pooling operation with a kernel size of two is performed, and then multiplied by the weight coefficient
$k^{2}$. The $k^{2}$ is the weight coefficient of training multi-channel feature maps in the network.

### 4.2. Multi-Scale Convolutional Neural Network Parameters

Before starting training, the parameters of the convolutional layer and the fully connected layer are initialized to a normal distribution, with an average value of 0 and a standard deviation of 1. The specific parameters and outputs of each layer of the multi-scale convolutional neural network are shown in Table 1.

### 4.3. Calculation Process

In the process of the forward propagation and back propagation of the multi-scale convolutional neural network, the specific operation process is shown in Formulas (4)~(7).
(4)$$\mathrm{sigmoid}(x)=\frac{1}{1+e^{-x}}$$

Formula (4) is the sigmoid activation function used in the neural network, and x is the input feature data of the activation function.
(5)$$X_{i+1}=\sum_{i=0}^{2}\sigma \left[\mathrm{Maxpooling}\left(k^{1}\left(W_{i}^{1}X_{i}+b_{i}^{1}\right)\mathrm{cat}\left(k^{2}\left(W_{i}^{2}X_{i}+b_{i}^{2}\right)\right)\mathrm{cat}\left(X_{i}\right)\right)\right]$$

In Formula (5), $X_{i+1}$ is the feature tensor value of the latent space output by the i + 1 block module, and σ is the sigmoid activation function. Max-pooling is the dimensionality reduction operation, $k^{1}$ is the weight parameter of the single-channel feature map, and $W_{i}^{1}$ is the neuron weight parameter value contained in the convolutional single-channel feature layer in the i block module. $b_{i}^{1}$ is the neuron bias parameter value contained in the convolutional single-channel feature layer in the i block module. $k^{2}$ is the weight parameter of the convolutional multi-channel feature layer. $W_{i}^{2}$ is the neuron weight parameter value contained in the convolutional multi-channel feature layer in the i block module, and $b_{i}^{2}\;$ is the neuron bias parameter value contained in the convolutional multi-channel feature layer in the i block module. High-dimensional latent features are concatenated in the channel dimension by using “cat”.
(6)$$Y_{j+1}=\sum_{j=0}^{1}\sigma \left(W_{j+1}Y_{j}+b_{j+1}\right),\;\mathrm{where}{\;Y}_{0}\;\mathrm{is}\;X_{3}\;\mathrm{in}\;(5)$$

In Formula (6), $Y_{j+1}$ is the output feature tensor value of the fully connected layer of j + 1 in the fc module. $W_{j+1}\;\mathrm{and}\;b_{j+1}$ are the weight parameter and bias parameter of the neuron of the j + 1 layer in the fc module, respectively, and $\;Y_{2}$ is the final output value of the network.
(7)$$\mathrm{MultiLabelSoftMarginLoss}=-\frac{1}{n}\sum (y_{n}\times \ln\sigma (x_{n})+(1-y_{n})\times \ln\left(1-\sigma \left(x_{n}\right)\right))$$

In Formula (7), $y_{n}$ is the true label value of n samples, $x_{n}\;$ is the network prediction value, and σ is the sigmoid activation function.

In the process of backpropagation, the gradient descent method is used to find the partial derivative of the loss function to w, b, and k, which are $\frac{\partial L}{\partial w}$,$\;\frac{\partial L}{\partial b}$, and $\;\frac{\partial L}{\partial k}$. W, b, and k are optimized by using the Adam optimizer with a learning rate of 0.001.

## 5. Experiment and Results

### 5.1. Object Attribute Strength Level Recognition Experiment

In this paper, PyTorch, a deep learning library based on Python, is used to build a neural network, and all haptic samples in the NumaTac haptic data set are trained and tested. For some deep networks, as resnet-152, the 23 in the data length (23 × 300 × 2) cannot carry out a very deep convolution operation. Therefore, considering the existing networks and our own needs, the multi-scale convolutional neural network is compared with a variety of models, which are mnasnet1_0 [39], resnet18 [40], shufflenet_v2 [41], and mobilenet_v2 [42]. The four models used in this paper are mature and advanced. They have been tested and the rationality of their network structure fully proved. The different evaluation criteria of accuracy, precision, recall, and f1_score are shown in Table 2.

### 5.2. Analysis of Results

In Table 2, the effects of the five models under the four evaluation criteria on the elasticity and hardness strength levels of 1–10 are counted. The four evaluation criteria are accuracy, precision, recall, and f1_score. 

#### 5.2.1. Accuracy

The accuracy is the ratio of the number of correctly classified samples to the total number of samples. It represents the overall prediction degree. Only when the actual value is completely consistent with the predicted output is the predicted output considered to be accurate. As can be seen from Table 2, the accuracy of the multi-scale convolutional neural network with regard to the elasticity strength level of objects is 9%, 4%, 5%, and 1%, respectively, higher than mnasnet1_0, resnet18, shufflenet_v2, and mobilenet_v2. The accuracy of the multi-scale convolutional neural network concerning the hardness strength level of objects is 14%, 14%, 10%, and 5%, respectively, higher than mnasnet1_0, resnet18, shufflenet_v2, and mobilenet_v2. This means that the difference of the multi-scale convolutional neural network between the output and the real value is the smallest, and the prediction effect of the multi-scale convolutional neural network is better in the object prediction process.

#### 5.2.2. Precision

The precision is the ratio of all the predicted positive samples to actually positive samples in the prediction result. Table 2 shows the precision of the sample strength level of all models. In fact, the worst precision of the multi-scale convolutional neural network only achieves 80% at elasticity strength level and 87% at hardness strength level, but it is still higher than other models, that is, the precision of mansnet1_0 at the ninth elasticity strength level is 35%, and the precision at the third hardness strength level is 35%. The precision of resnet18 is 65% at the second elasticity strength level and 39% at the seventh hardness strength level. The precision of shufflenet_v2 at the first elasticity strength level is 76%, and the precision at the second hardness strength level is 63%. The precision of mobilenet_v2 at the third elasticity strength level is 71%, and the precision at the sixth hardness strength level is 68%. The above proves that the prediction of the multi-scale convolutional neural network for all strength levels is more stable and can be maintained at a higher recognition rate.

#### 5.2.3. Recall

The recall is the ratio of the actual positive samples to predicted positive samples. As can be seen from Table 2, the worst recall of the multi-scale convolutional neural network is 74% at the third elasticity strength level and 81% at the eighth hardness strength level. Meanwhile, the worst recall of mansnet1_0 at the second elasticity strength level is 77%, and at the eighth and ninth hardness strength level is 75%. Resnet18 has a worst recall of 73% at the tenth elasticity strength level and 75% at the third hardness strength level. Shufflenet_v2 has a worst recall of 71% at the eighth elasticity strength level and 75% at the seventh hardness strength level. The worst recall of mobilenet_v2 at the fourth elasticity strength level is 69% and, at the sixth hardness strength level, is 77%. The recognition rate of the other nine levels of the multi-scale convolutional neural network remains at a higher level than other models.

#### 5.2.4. F1_Score 

F1_score is an indicator that comprehensively considers the precision and the recall. Therefore, in order to measure the quality of the model in a more comprehensive and balanced manner, the quality of the model is evaluated by the harmonic average of the precision and the recall. As can be seen from Table 2, f1_score of the multi-scale convolutional neural network is 98% at the fifth elasticity strength level and 99% at the tenth hardness strength level, which are higher than the other models. Meanwhile, the f1_score of mansnet1_0 is 89% at the fourth elasticity strength level and 91% at the seventh hardness strength level. The f1_score of resnet18 is 94% at the first elasticity strength level and 89% at the ninth hardness strength level. The shufflenet_v2 has an f1_score of 89% at the fourth elasticity strength level and 94% at the ninth hardness strength level. The f1_score of mobilenet_v2 is 97% at the fifth elasticity strength level and 97% at the tenth hardness strength level. The above proved that the multi-scale convolutional neural network’s predictions for all strength levels remain at a higher recognition rate.

To summarize, compared with other models, the multi-scale convolutional neural network is more accurate and it has a significant improvement in terms of the recognition rate. From the point of view of precision, in the prediction results of the positive examples in the predicted value, the true value has a higher proportion of positive examples. The prediction effect of positive examples in the true value is better from the perspective of the recall. Finally, the performance of the recognition effect is more stable and the recognition rate is higher from the perspective of the f1_score. It can be concluded that the recognition of attribute strength level of the multi-scale convolutional neural network performs better.

## 6. Discussion

There are abundant tactile sensors on the human palm, so it is an interesting idea to carry tactile sensors on the robotic arm. By designing the control and algorithm of the robotic arm, the robot’s intelligent perception level of the target can be enhanced. 

The purpose of this paper is to enable robots to understand the property strength levels of objects in a more detailed manner. To this end, this paper uses the Kinova manipulator and the NumaTac tactile sensor to establish a data set of tactile properties of objects. The signals in the tactile samples of objects are force signals and vibration signals that change over time. These two signals can reflect the properties of elasticity and hardness of objects. The data set is a balanced data set for the two properties of sample elasticity and strength.

In this paper, a multi-scale convolutional neural network method for recognizing the attribute strength levels of objects is proposed and, in order to verify the effectiveness of the method, the method was trained and tested using a haptic data set. Compared with the comparison model, experiments show that this method can better extract the tactile data features of objects and can identify the elasticity strength level and hardness strength level of the objects. We believe that improving the speed of multi-scale convolutional neural networks is a challenge. In the current research trend, the speed of the network is a key issue, which is also the focus of our next research. On the basis of this work, using robots to identify objects online is a good application direction, and this part of the work may be carried out in future studies.

## 7. Conclusions

In this paper, a robotic arm platform is used. The platform includes a robotic arm and a robotic gripper. Each of the two fingers on the gripper is equipped with a NumaTac tactile sensor. Through this operating platform, we established a tactile data set of object attribute strength levels, selected two properties of the object (elasticity and hardness) for analysis, and proposed a multi-scale CNN to identify the object’s elasticity strength level and hardness strength level. The experimental results show that the network performs better in terms of prediction accuracy and prediction stability.

In future work, we propose to focus on the relationship between tactile dimensions and a more delicate tactile attribute strength level classification method to identify the objects’ strength levels. Besides, compared with some well-known data sets, our data set is not large and we plan to expand the data set in our future work. Finally, how to apply it to practical robotics applications, such as product identification in the industrial field, is also a problem that should be considered.

## Figures and Tables

**Figure 1 sensors-22-01908-f001:**
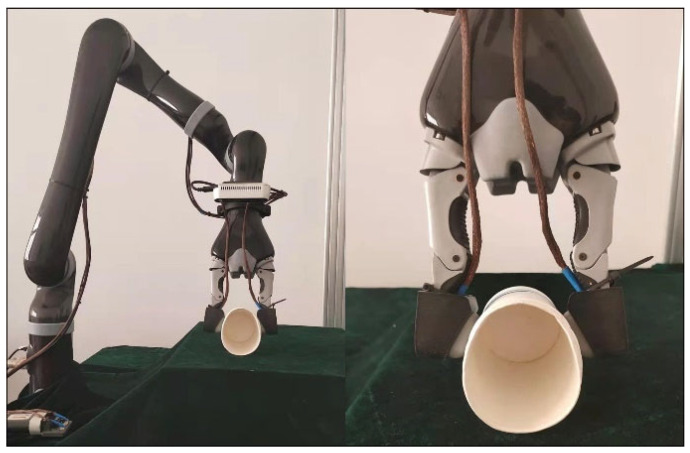
Robot operating platform that is picking up a paper cup.

**Figure 2 sensors-22-01908-f002:**
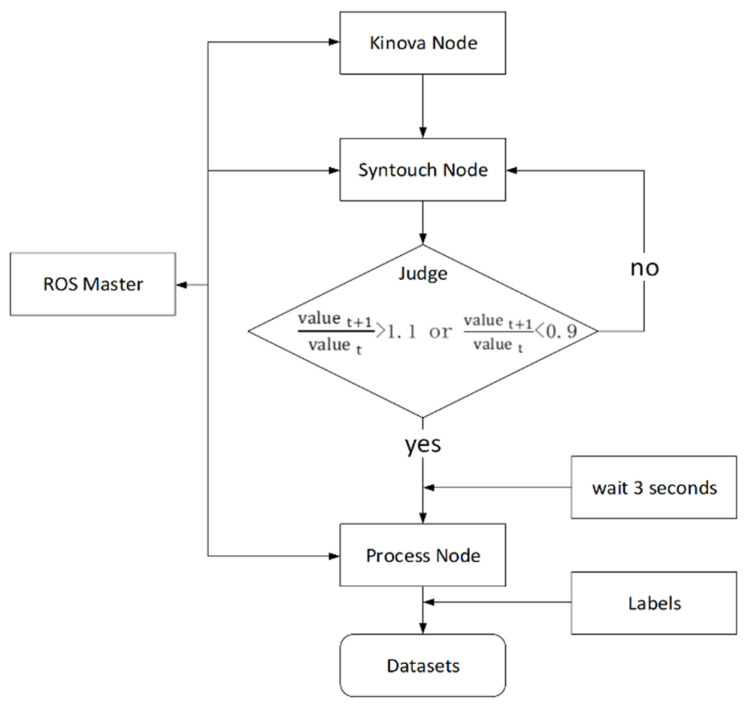
ROS graph of the data acquisition platform.

**Figure 3 sensors-22-01908-f003:**
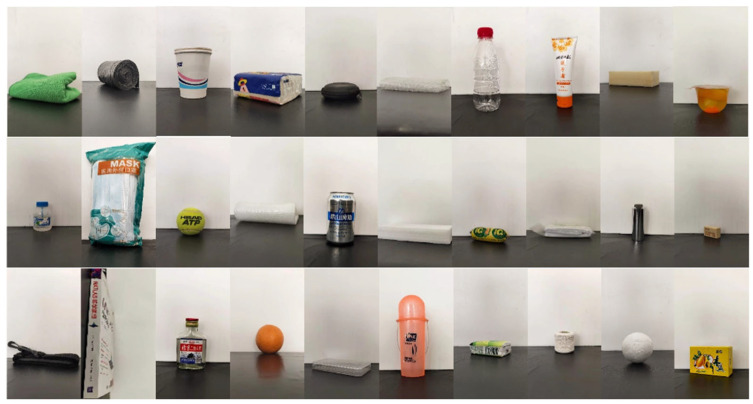
A total of 30 objects in the NumaTac haptic data set (from left to right and from top to bottom; the objects are a towel, garbage bag, paper cup, medium draw paper, earphone box, bubble wrap, mineral water bottle, hand cream, square sponge, jelly, glue, mask, tennis, double layer foam, can, square foam, ham sausage, triangular bandage, metal column, rubber, black bandage, book, glass bottle, orange, plastic box, dental cylinder box, small draw paper, white thread group, foam ball, and soap box).

**Figure 4 sensors-22-01908-f004:**
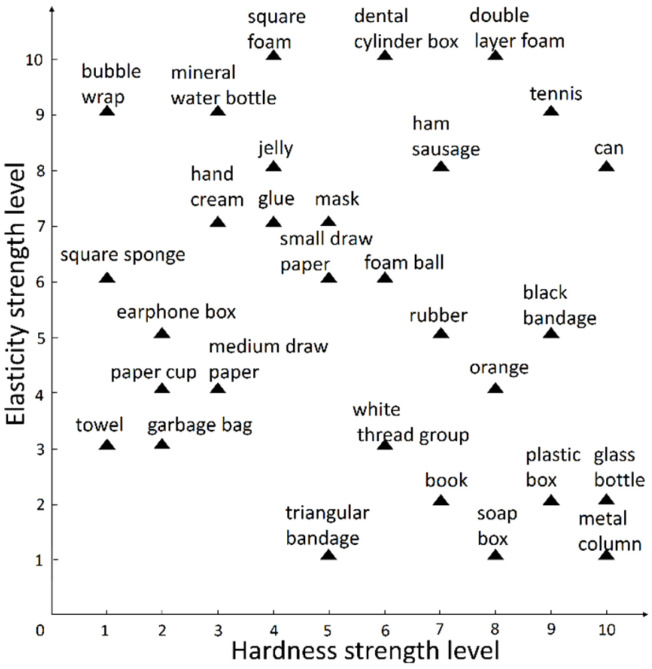
Horizontal graph of the elasticity and hardness strength level of 30 objects in the NumaTac haptic data set.

**Figure 5 sensors-22-01908-f005:**
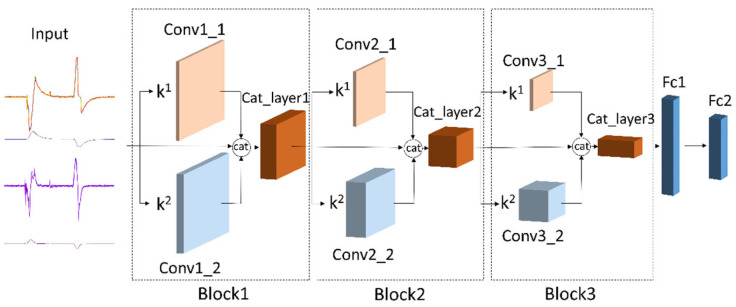
Multi-scale convolutional neural network architecture.

**Table 1 sensors-22-01908-t001:** Parameters of each layer of the multi-scale convolutional neural network.

Layers		Parameters	Next Operation	Output
		Size, Stride, Padding		
Input	Input max-pooling	2 × 2, 0, 0		23 × 300 × 2 11 × 150 × 2
Conv1_1	conv max-pooling	3 × 3 × 1, 1, 1 2 × 2, 0, 0	BN	23 × 300 × 1 11 × 150 × 1
Conv1_2	conv max-pooling	3 × 3 × 5, 1, 1 2 × 2, 0, 0	BN	23 × 300 × 5 11 × 150 × 5
Cat_layer1			Leaky_relu	11 × 150 × 8
Conv2_1	conv max-pooling	3 × 3 × 1, 1, 1 2 × 2, 0, 0	BN	11 × 150 × 1 5 × 75 × 1
Conv2_2	conv max-pooling	3 × 3 × 15, 1, 1 2 × 2, 0, 0	BN	11 × 150 × 15 5 × 75 × 15
Cat_layer2			Leaky_relu	5 × 75 × 24
Conv3_1	conv max-pooling	3 × 3 × 1, 1, 1 2 × 2, 0, 0	BN	5 × 75 × 1 2 × 37 × 1
Conv3_2	conv max-pooling	3 × 3 × 32, 1, 1 2 × 2, 0, 0	BN	5 × 75 × 32 2 × 37 × 32
Cat_layer3			Leaky_relu	2 × 37 × 57
Fc1		(4218,512)	BN	512
Fc2		(512,20)	BN	20

**Table 2 sensors-22-01908-t002:** Accuracy, precision, recall, and f1_score of elasticity and hardness strength levels of the five models, and the best or worst result shown in bold.

Net	Level	Elasticity Level_Score (%)				Hardness Level_Score (%)			
		accu	prec	reca	f1_score	accu	prec	reca	f1_score
Multi-scale convolutional neural network	1	**90**	95	77	85	**95**	95	89	92
	2		85	98	91		**87**	97	92
	3		**80**	**74**	77		93	95	94
	4		85	89	87		93	99	95
	5		98	99	**98**		96	100	98
	6		98	82	89		99	95	97
	7		94	96	95		95	96	95
	8		82	97	89		98	**81**	89
	9		93	95	94		97	99	98
	10		95	94	94		99	99	**99**
Mnasnet1_0	1	**81**	98	79	87	**81**	99	83	91
	2		98	**77**	87		100	81	90
	3		95	78	86		**35**	99	52
	4		98	81	**89**		99	82	90
	5		99	79	88		99	76	86
	6		99	79	88		100	78	88
	7		99	77	86		99	83	**91**
	8		99	78	87		98	**75**	85
	9		**35**	99	52		100	**75**	85
	10		100	80	89		99	81	89
Resnet18	1	**86**	98	91	**94**	**81**	100	76	86
	2		**65**	99	78		83	89	86
	3		94	75	83		93	**75**	83
	4		83	91	87		96	77	86
	5		80	96	87		98	79	87
	6		99	81	89		100	77	87
	7		93	83	87		**39**	99	56
	8		80	84	82		100	79	88
	9		87	82	85		100	81	**89**
	10		100	**73**	85		98	81	88
Shufflenet_v2	1	**85**	**76**	81	78	**85**	88	86	87
	2		85	93	89		**63**	90	74
	3		90	81	86		90	81	85
	4		88	91	**89**		86	82	84
	5		85	89	87		86	93	90
	6		88	87	88		94	85	90
	7		82	84	83		83	**75**	79
	8		79	**71**	75		89	82	85
	9		91	83	87		93	95	**94**
	10		85	88	86		93	85	89
Mobilenet_v2	1	**89**	96	87	91	**90**	90	95	92
	2		97	95	96		99	91	95
	3		**71**	95	81		96	**77**	86
	4		97	**69**	83		99	88	93
	5		97	97	**97**		97	85	90
	6		99	83	91		76	95	85
	7		97	90	93		**68**	97	80
	8		73	92	81		97	93	95
	9		94	90	92		98	87	92
	10		97	93	95		99	95	**97**

## Data Availability

https://github.com/yuguoqi-learner/identifying-the-strength-level.git.

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
