# Peer review of "Identifying the Strength Level of Objects’ Tactile Attributes Using a Multi-Scale Convolutional Neural Network"

_sensors, 2022, doi:10.3390/s22051908_

Round 1

Reviewer 1 Report

In this paper, the author proposes a multi-scale convolutional neural network to identify the strength level of the elasticity and hardness of 30 objects. The experimental results show that the network performs better in terms of prediction accuracy and prediction stability.This manuscript could be improved considering the following comments: 

Point 1: Authors should check the English expression of this paper, especially the abstract and section 5.2, so that the goals and results of the study can be better understood. 

Point 2: Introduction section could include more discussion about the innovation of the article and the main advantages of the proposed Multi-scale Convolutional Neural Network in comparison with others reported in the literature.

Point 3: Quality of the figure 4 should be improved.

Point 4: Why did the authors choose those four models as a comparison In section 5.1?

Point 5: Which are the main limitations or challenges of the proposed Multi-scale Convolutional Neural Network?

Author Response

Dear reviewer:

We have carefully considered your comments and replied to them one by one. The revised part is marked with "track changes" . In the following, the answer is abbreviated as A.

Point1: Authors should check the English expression of this paper, especially the abstract and section 5.2, so that the goals and results of the study can be better understood.

A1: We have revised the English expression of this paper, which can be viewed in the revised manuscript.

Point2: Introduction section could include more discussion about the innovation of the article and the main advantages of the proposed Multi-scale Convolutional Neural Network in comparison with others reported in the literature.

A2: First of all, this paper established a tactile-based object attribute data set. Samples in the data set contain the elasticity and hardness characteristic information of objects, and each attribute is divided into ten levels to describe the object more delicately. Compared with other tactile data set in this field, the unique contribution of the tactile data set established in this paper is to solve the problem of lack of relevant data about the intensity difference of object attributes, and add information about the intensity difference of object attributes in the data set. The data set lays a foundation for more refined cognition and operation of objects through robot touch. Second, a method for identifying the attribute strength level of a convolutional neural network by using multi-scale features is proposed, which can identify the difference in the strength of elasticity and hardness of different objects in the data set. The main contribution of this method is that in each feature extraction process, the multi-scale features of the original data are comprehensively superimposed, which makes the network understand the data more comprehensively. Compared with other methods, the unique advantage of this method is that the integrity and sufficiency of information are fully considered in each step of feature extraction, which means that this method is sensitive to data changes. The elasticity strength is regarded as the strength of the feedback to the experimenter when the object returns to its original state after deformation, and the hardness is regarded as the difficulty of the deformation of the object under the same force.

Point3: Quality of the figure 4 should be improved.

A3: We have improved the quality of figure 4

Point 4: Why did the authors choose those four models as a comparison In section 5.1?

A4: Firstly, the four models used in this paper are mature and advanced. They have been tested in other studies, which fully proves the rationality of their network structure. Secondly, for some deep networks, the 23 in the data length (23×300×2)cannot carry out very deep convolution operation. Therefore, considering the existing networks and our own needs, we use the four models in this paper for comparison.

Point 5: Which are the main limitations or challenges of the proposed Multi-scale Convolutional Neural Network?

A5: We believe that reducing the parameters of multi-scale convolutional neural networks is a challenge. In the current research trend, the speed of the network is a key issue, which is also the focus of our next research.

Reviewer 2 Report

This paper proposed a tactile-based object attribute data set, which contain the elasticity and hardness characteristics of different objects. The authors also developed a method to identify the attribute strength level of a convolutional neural network by using multi-scale features. The authors defined the elasticity strength as the feedback to the person when the object returns to its original state after deformation, and the hardness as the difficulty of the deformation of the object under the same force. With this method and the defined features, the difference in the strength of elasticity and hardness of different objects could be identified in the data set. The motivation of this dataset is that the binary label in previous works does not include the knowledge of the strength level of the object's delicate attribute. The authors clearly provided detailed procedures of acquiring the data. The paper was also well organized and written clearly. Please find below some comments to help improve the manuscript.

  1. It would be helpful if the authors could compare with other tactile dataset and mark the unique contribution of this work to help understand what have been added on top of existing works.
  2. Are the tactile attributes such as the elasticity, hardness and other properties coupled together?
  3. Is there a way to quantitatively measure the elasticity and other properties as the ground truth?
  4. Is this dataset available online so that other people can potentially use it for benchmark test?
  5. Figure 3 provided 30 objects to be tested and included in the data set. Why are these objects selected?
  6. Figure 4 is very unclear with a low resolution. It is difficult to see contents.
  7. The authors mentioned about some application of the data set. Could you provide some concrete examples of how to use this data set?
  8. What are some of the limitations of the data set such as data set size, etc.? How will you improve?

Author Response

Dear reviewer:

We have carefully considered your comments and replied to them one by one. The revised part is marked with "track changes" .In the following, the answer is abbreviated as A.

Point 1: It would be helpful if the authors could compare with other tactile dataset and mark the unique contribution of this work to help understand what have been added on top of existing works.

A1: Compared with other tactile data set in this field, the unique contribution of the tactile data set established in this paper is to solve the problem of lack of relevant data about the intensity difference of object attributes, and to add information about the intensity difference of object attributes in the data set. The data set lays a foundation for more re-fined cognition and operation of objects through robot touch.

Point 2:.Are the tactile attributes such as the elasticity, hardness and other properties coupled together?

A2: Yes, the elastic and hardness properties of each object are coupled together.

Point 3: Is there a way to quantitatively measure the elasticity and other properties as the ground truth?

A3: It is possible to use special instruments to measure the specific attribute strength value of objects, but this paper focuses on the difference between attribute strength. Therefore, as long as the tactile attribute strength level can reflect the difference, it can meet the research needs in this paper. We plan to measure the specific attribute strength value of objects in the next research, and the specific work needs to be carried out in the future.

Point 4:Is this dataset available online so that other people can potentially use it for benchmark test?

A4: Yes, we plan to publish this data set in the future.

Point 5: Figure 3 provided 30 objects to be tested and included in the data set. Why are these objects selected?

A5: Firstly, the 30 objects contain a variety of shapes and materials, which can meet our needs to realize attribute strength difference recognition. Secondly, we consulted advanced papers and materials, and finally we determined all the objects to be tested.

Point 6: Figure 4 is very unclear with a low resolution. It is difficult to see contents.

A6: We have revised figure 4.

Point 7: The authors mentioned about some application of the data set. Could you provide some concrete examples of how to use this data set?

A7: Because the data set contains vibration and pressure signals, we can use this data set to apply to product identification in the industrial field.

Point 8: What are some of the limitations of the data set such as data set size, etc.? How will you improve?

A8: Compared with some well-known data sets, our data set is not large, and we plan to expand the data set in our future work.

This manuscript is a resubmission of an earlier submission. The following is a list of the peer review reports and author responses from that submission.

Round 1

Reviewer 1 Report

1. First of all, this manuscript contains numerous typographical and grammatical errors. The authors should check your manuscript carefully supported by a native English speaker or a related service before submission. It's hard to read the current manuscript.

2. Subsequently, the figures and tables are low resolutions, especially the texts and numbers, which are also difficult to read.

3. Essentially, the novelty of this study is not readily apparent. In addition, the survey of related studies is insufficient.

4. The authors should cite the literature for the CNNs used in your study. Did you use LeNet-5? Although this manuscript did not mention the backbone of CNNs, I estimated it from Figure 6. Currently, LeNet-5 is an old backbone. For improving accuracy, You need to use the latest backbones such as Xception, ResNet, Inception, InceptionResNet, Inception, MobileNet, DenseNet, etc.

5. Comparison experiments should be conducted with state-of-the-art methods outside of CNN derivatives.

6. The set values of the network parameters are unclear.

7. The consideration of effective values of accuracy is insufficient.

8. What dimensions are the input features to the network as depicted in Figure 6?

9. What was the basis for determining the ratio as described in line 331?

10. A list of abbreviations is required in the last part of the manuscript.

11. Qualitative expressions such as "good effect" are unnecessary.

Reviewer 2 Report

The paper presents a work focused on tactile attribute recognition using CNN. In concrete the authors propose to use their own dataset to train the presented neuronal network method. Although the topic discussed in this paper is of interest for the robotic manipulation task community, this presents serious flaws.

First, the authors must review English language and style. Current manuscript state is difficult to follow because multiple typos, grammar error and incorrect expression, in addition figures and tables have low quality. Moreover, the authors use figures from other sources, for example figure 2 is originally shown in

According to the authors, now a days there is a lack of disposal dataset in the literature related with tactile information, due to that, the authors present its own dataset. Because it is a new dataset, the authors might make public this dataset. But using the information presented in the paper, it will be difficult to use, I miss relevant information like, how is the raw information code (how many channels of information have this tactile sensor of?), other questions like is the dataset balanced are not responded. More questions about the dataset, is there motion of the robot arm during the manipulation task, i.e., is it just a grasp / hold / ungrasp task? Or during the grasp and ungrasp does the arm realize any other movement? If that the case the author must provide that information.

Related with the labeling method presented here, it is not clear how the authors bin the raw information neither how they solve the multi-label problem. In addition it is not clear why the authors treated the problem of strength like a classification problem instead of a regression one.

Reviewer 3 Report

The manuscript titled ‘Recognition of Tactile Attribute Strength and Category Using 2 Convolutional Neural Network’ aims to enable the robot not only to recognize the tactile properties of objects, but also to understand the strength of the properties of objects more delicately,  and then to recognize the categories of objects. This paper uses Kinova manipulator and NumaTac tactile sensor to establish the haptic dataset of the intensity of object attributes.The haptic sample contains force signal and vibration signal,  and the matrix label contains the elastic strength and hardness strength information of the object attribute. Then, a convolutional neural network(CNN) based object attribute strength and category recognition algorithm is proposed, which is used to identify the elastic strength, hardness strength and object category of the object in the haptic dataset.

The work is interesting and innovative, the results is encouraging;only some minor aspects should be clarified by the authors:

  • Abstract: Why is it divided in sections and not in a unique and continuous form?
  • L40: Whether why uppercase?
  • L120 why The uppercase?
  • Can you explain the coefficients 12.94 and 0.12 in eqs. (1) and (2)?
  • L160and following you should add some references about ROS nodes
  • L212 why Squeeze uppercase?
  • L238 3S, why S uppercase?
  • L244 two commas and after In uppercase, please check typing errors in all the manuscript.
  • L248 can you explain the normalization carried out in Eq.(3)?
  • L276 why One is uppercase?
  • L331, do you think that the chosen ratio of 4:1 for training data and set data is acceptable for the real applications? What happened by brutally decreasing this ratio?
  • Could you represent the Euclidean distances depicted in Figs 8 and 9 in two tables for permitting a clearer analysis of the results?

Reviewer 4 Report

The FIGURES and TABLES must be improved, there are many sentences that we cant understand what is intended. Despite the interest of the application, the authors did not develop any algorithm or solution.